# Enhancing Sugarcane Yield and Sugar Quality through Optimal Application of Polymer-Coated Single Super Phosphate and Irrigation Management

**DOI:** 10.3390/plants12193432

**Published:** 2023-09-29

**Authors:** Muhammad Sajid, Muhammad Amjid, Hassan Munir, Mohammad Valipour, Fahd Rasul, Aka Khil, Mashael Daghash Alqahtani, Muhammad Ahmad, Usman Zulfiqar, Rashid Iqbal, Muhammad Fraz Ali, Iqra Ibtahaj

**Affiliations:** 1Department of Agronomy, University of Agriculture Faisalabad, Faisalabad 38040, Pakistan; saajiduaf@gmail.com (M.S.); amjidm70@gmail.com (M.A.); hmbajwauaf@gmail.com (H.M.); drfahdrasul@gmail.com (F.R.); ahmadbajwa516@gmail.com (M.A.); 2Department of Engineering and Engineering Technology, Metropolitan State University of Denver, Denver, CO 80217, USA; 3Department of Biology, College of Science, Princess Nourah bint Abdulrahman University, P.O. Box 84428, Riyadh 11671, Saudi Arabia; 4Department of Agronomy, Faculty of Agriculture and Environment, The Islamia University of Bahawalpur, Bahawalpur 63100, Pakistan; rashid.iqbal@iub.edu.pk; 5College of Agronomy, Northwest A & F University, Xianyang 712100, China; frazali15@gmail.com; 6Department of Botany, University of Agriculture Faisalabad, Faisalabad 38040, Pakistan

**Keywords:** sugarcane, yield, cane sugar recovery %, irrigation, polymer-coated SSP

## Abstract

The judicious use of crop input is of prime importance for achieving a considerable output with a low-cost input. A two-year field experimentation was executed to assess the effect of varying polymer-coated single super phosphate (SSP) regimes on the yield and quality of sugarcane under differential water regimes. A two-factor study was executed under a randomized complete block design with a split-plot arrangement. The CPF-249 sugarcane variety was planted during the 2019–2020 period and the 2020–2021 period. The experiment consisted of four levels of polymer-coated SSP, i.e., control, 90, 110, and 130 kg ha^−1^, and three water regimes, which consisted of a number of irrigations, i.e., 18 irrigations, 15 irrigations, and 12 irrigations. Moreover, the water regimes were kept in the main plot, whereas the polymer-coated supplement was allocated in a subplot and replicated thrice. The data on the yield components and sugar-related traits were recorded during both years of study, and the treatment means were differentiated using an LSD test at a 95% confidence interval. Summating the findings of this study, a significant variation was revealed under the subject levels of both factors. Statistically, a 110 kg ha^−1^ polymer-coated SSP dose, along with 18 irrigations, declared the highest millable canes, stripped cane yield, and unstripped cane yield, followed by the 130 kg ha^−1^ treatment. Additionally, the highest pol% and cane sugar recovery % were recorded under 12 irrigations along with 130 kg ha^−1^ during both years. Similarly, the °Brix value was also significantly affected by 12 irrigations when 110 kg ha^−1^ of polymer-coated SSP was used. The unstripped cane yield had a strong positive correlation with the stripped cane yield, millable canes, and the number of internodes. Moreover, the commercial cane sugar % resulted in a strong positive correlation with the pol%, whereas the cane sugar recovery % revealed a strong positive correlation with the pol% and commercial cane sugar %.

## 1. Introduction

Sugarcane (*Saccharum officinarum* L.) plays a vital role on a global scale, making up roughly 85% of the sugar consumption across the world. In terms of sugarcane production, Pakistan holds the ninth position globally and ranks fifth in terms of the total acreage dedicated to cane cultivation. In the period of 2022–2023, Pakistan achieved a sugarcane yield of 91.11 million tons, marking a 2.8% growth compared to the previous year’s production of 88.65 million tons [1]. Sugarcane is a significant industrial plant that is cultivated in tropical and subtropical zones worldwide. Its primary purpose is the production of sugar, contributing to over 70% of the global sugar supply. Moreover, it serves as a dependable and renewable resource for bioenergy. Its cultivation spans across more than 24.9 million hectares in approximately 80 countries, resulting in a substantial production volume of 174 million tons [2].

Droughts stand as pivotal constraints that diminish the yield potential of numerous crops. The quantity of water utilized by a crop maintains a close correlation with its photosynthetic activity, generation of dry matter, and eventual yield, and it is a relationship that is widely observed in various plant species [3]. Particularly, the tillering and grand growth stages, constituting the crucial formative phase of sugarcane, have been pinpointed as the pivotal periods necessitating ample water supply. This is predominantly due to the fact that during this phase, a substantial 70–80% of the total cane yield is generated [4]. The indicators of plants grappling with drought stress encompass noteworthy morphological changes such as reductions in the leaf area, stunted root growth, and the closure of stomatal pores. Notably, droughts occurring during the formative phase of sugarcane resulted in a decrease in dry matter production, ranging from 46% to 61% [5].

The capacity of sugarcane varieties to endure drought stress varies, leading to varying degrees of yield reduction. Sugarcane plants that exhibit tolerance display elevated levels of photosynthetic activity when contrasted with their susceptible counterparts [6]. Typically, drought conditions lead to a decline in photochemical efficiency (PSII) and the cultivar’s capability to uphold a heightened level of Fv/Fm—a parameter indicative of radiation utilization efficiency and carbon assimilation. This phenomenon has emerged as a promising technique for the selection of drought-tolerant cultivars [7]. The reaction to drought stress is primarily orchestrated by plant hormones, specifically abscisic acid, auxin, gibberellic acid, cytokinin, brassinosteroid, jasmonic acid, ethylene, and strigolactone [8]. Similarly, adeptly managing freezing stress (caused by cold-induced water stress) through effective osmotic adjustment capabilities and a robust antioxidant system presents encouraging avenues for the genetic enhancement of cold tolerance in agricultural crops [9].

The use of superabsorbent polymers might be an efficient and environmentally friendly approach to enhance the functional capacity of rainfed and irrigated cropping systems, increase crop and water productivity, enhance farm income, ensure food security, conserve natural resources, and ultimately alleviate rural poverty. The BSPs, as water-retaining materials, can store a huge quantity of water and nutrients and release them slowly when required by plants. A number of studies have previously documented the beneficial effects of BSPs in enhancing crop growth and performance [10,11,12,13,14,15]. Nano clay polymer composite superabsorbent improves water absorption and water retention capabilities and can effectively be applied in rainfed agriculture to mitigate water stress [16]. Mazloom et al. [17] reported that soil receiving 0.6% lignin hydrogel produced a significantly greater maize biomass and relative leaf water content under severe water-limited conditions.

Due to the presence of many hydrophilic groups and three-dimensional cross-linked network structures, polymer water retention agents exhibit huge water absorbency and strong water retention capacity even under adverse conditions [13]. The superabsorbent polymer application increased both the water holding capacity and available water while reducing runoff and evaporation losses [18,19] enough to meet the water requirement of normal plant growth [20,21]. Starch-modified poly (acrylic acid)-based hydrogel showed the highest water absorption when 0.25% hydrogel was added, and the water holding capacity was enhanced by 120% [14]. Moreover, hydrogels such as polyacrylamide have the potential to contribute to water use efficiency under abiotic stress conditions, and up to 95% of the residual acrylamide was degraded within 30 days, indicating negligible environmental persistence [15]. Thombare et al. [22] synthesized a novel hydrogel by grafting guar gum with acrylic acid and cross-linking it with ethylene glycol di methacrylic acid and found that it absorbed up to 800 mL of water per gram and significantly improved the soil porosity, moisture absorption, and retention capacity compared with the control. Tao et al. [23] indicated that carbohydrate-based polymers improved the root length, shoot length, total biomass, germination potential, and germination rate, indicating that it was not toxic to plants. Li et al. [11] stated that an acrylamide–potassium acrylate copolymer significantly increased plant growth, chlorophyll contents, photosynthesis efficiency, and the relative water content and enhanced the activities of antioxidant enzymes in Areca catechu L. under severe water deficit conditions. Taking into consideration the above-mentioned facts, this study was arranged to assess the effect of polymer-coated SSP on sugarcane yield and quality traits by employing different irrigation regimes under the semi-arid conditions of Faisalabad.

## 2. Results

### 2.1. Description of Main Effects

The analyzed data indicate that the main effects of all parameters were significant, excluding the fiber% under both factors across the years (Table 1); the °Brix value was also insignificant under varying water levels during the 2020–2021 period (Table 2). However, the highest number of internodes during the 2019–2020 period were counted under 18 irrigations, followed by 15 irrigations, and the lowest number of internodes were collected where 12 irrigations were imposed (Table 1). Conversely, a similar trend was also documented during the 2020–2021 period. On the other hand, during both years of study, 110 kg ha^−1^ of polymer-coated SSP resulted in the highest number of internodes, and the lowest value was observed where no polymer-coated SSP fertilizer was used (Table 1). Similarly, during the 2019–2020 period, statistically, the highest pol% was calculated with the application of 12 irrigations, followed by 15 irrigations, and the lowest percentage was declared at under 18 irrigations, whereas during the 2020–2021 period, the highest pol% was recorded by employing 18 irrigations, followed by the rest of the water levels that were statistically similar. Nonetheless, in the case of polymer-coated supplements, 130 kg ha^−1^ significantly increased the pol percentage in the 2019–2020 period, which was statistically similar to 110 kg ha^−1,^ and the lowest value was calculated under the control treatment. In a comparison of the 2020–2021 period with the previous year, 130 kg ha^−1^ of polymer-coated SSP resulted in the highest poll percentage, followed by 110 kg ha^−1^ and 90 kg ha^−1^, and the minimum value was revealed where no polymer-coated SSP was applied (Table 2). The commercial cane sugar (CCS) percentage during the 2019–2020 period was the highest when 18 irrigations were applied, statistically similar results were found with 15 irrigations, and the lowest CCS% was documented where no fertilizer was applied. However, in the 2020–2021 period the highest CSS% was recovered where 18 irrigations were applied, and the lowest percentage was calculated under 12 irrigations. Generally, the 2019–2020 period had the highest CCS% compared to the 2020–2021 period. In the 2019–2020 period, 130 kg ha^−1^ of polymer-coated SSP had the highest CCS% that was reflected in an identical percentage under 110 kg ha^−1^ and the control treatment, and the lowest value was permitted under 90 kg ha^−1^. Conversely, in the 2020–2021 period, statistically, the leading value was examined when the 130 kg ha^−1^ dose was applied, whereas the lowest value was exhibited where no polymer-coated SSP was used (Table 2).

### 2.2. Interactive Effects of Significant Traits

#### 2.2.1. Millable Canes and Stripped Cane Yield

As far as the interactive effect is concerned, it is evident from Figure 1 that there is a significant variation for millable canes and stripped cane yield in response to different polymer-coated SSP regimes under different water levels over the years. During the 2019–2020 period, statistically, the leading millable canes were harvested with the application of 110 kg ha^−1^ of polymer-coated SSP, keeping all water levels, followed by 130 kg ha^−1^, whereas the least millable canes were obtained where no polymer-coated SSP was applied. In addition, a similar trend was also exhibited in the 2020–2021 period, and the first year trial resulted in more canes compared to the second year. However, the stripped cane yield was also significantly influenced by the employment of subject factors, and in both years’ trials, the highest yield was collected from the crop using 110 kg ha^−1^ of polymer-coated SSP followed by 130 kg ha^−1^, whereas during both years, the lowest stripped cane yield was obtained when the control treatment was employed (Figure 1).

#### 2.2.2. Unstripped Cane Yield and °Brix

The interactive effects of the subject traits showed a significant response by keeping the fertilizer doses and water regimes (Figure 2). During the 2019–2020 period, the use of a polymer-coated SSP at 110 kg ha^−1^ significantly improved the unstripped cane yield where 18 irrigations, 15 irrigations, and 12 irrigations were applied followed by 130 and 90 kg ha^−1^ doses, and the lowest findings were observed in the control treatment. Overall, the lowest yield was recorded under 12 irrigationscompared to the others. In addition, during the 2020–2021 period, similar results were also declared under the studied factors. The data regarding the °Brix value in the 2019–2020 period showed that statistically, the highest °Brix value was measured under 130 kg ha^−1^ of polymer-coated SSP with 18 irrigations, and the lowest value resulted under the control treatment. However, under 15 irrigations and 12 irrigations, the emerged value was observed where 110 kg ha^−1^ of polymer-coated SSP was imposed, and the lowest °Brix value was noted where no polymer-coated fertilizer was applied (Figure 2).

#### 2.2.3. Cane Juice Purity % and Cane Sugar Recovery %

Regarding the subject explanation of targeted traits, the interactive effect between the water levels and fertilizer supplement exhibited a significant variation in both years of experimentation (Figure 3). During the 2019–2020 period, under the 18-irrigation treatment, 110 kg ha^−1^ resulted in a significant increase in the cane juice purity % that was statistically at par with 130 kg ha^−1^, whereas a low reading was observed where 90 kg ha^−1^ was used. In the case of 15 irrigations and 12 irrigations, the highest cane juice purity was observed under all supplements, whereas the lowest value was observed in the control treatment where no polymer-coated SSP was given. Similarly, the in 2020–2021 period, the application of 90 kg ha^−1^ of polymer-coated SSP along with 18 irrigations had the highest value that was similar to the rest of the treatments, and the lowest value was recorded under 110 kg ha^−1^. Under the 15-irrigation treatment, 130 kg ha^−1^ reflected the highest percentage, and a parity finding was seen with 90 kg ha^−1^, and the lowest purity percentage was calculated where no polymer-coated SSP was used. Similarly, 12 irrigations resulted in the highest purity percentage under all SSP regimes, excluding the control treatment, which showed the lowest cane juice purity percentage.

Moreover, the cane sugar recovery was significantly influenced by employing water levels and fertilizer regimes over the years. During the 2019–2020 period, 18 irrigations showed a significant increase with the use of 130 kg ha^−1^ and 110 kg ha^−1^ of polymer-coated SSP, and the least sugar recovery was recorded where 90 kg ha^−1^ was applied. Under 15 irrigations, 130 kg ha^−1^ of polymer-coated SSP exhibited the maximum sugar recovery percentage, whereas the least recovery was calculated where the control treatment was imposed. Similarly, with 12 irrigations, the highest sugar recovery was assessed when 130 kg ha^−1^ of polymer-coated SSP was applied, and the lowest percentage was found under 90 kg ha^−1^. However, under 18 irrigations, the polymer-coated SSP at 130 kg ha^−1^ significantly contributed to the cane juice purity when the crop was sown in the 2020–2021 period, and the lowest percentage was reflected under the control conditions. Correspondingly, 15 irrigations had the highest recovery percentage when the 130 kg ha^−1^ dose was used, and the lowest value was assessed in the control treatment. Inclusively, 12 irrigations had the highest recovery percentage compared to the other water levels. Moreover, the highest sugar recovery was assured from the 130 kg ha^−1^ and 110 kg ha^−1^ doses, whereas the minimum value was unveiled where no polymer-coated SSP was applied (Figure 3).

### 2.3. Correlation Explanation

A correlation analysis was executed to assess the relationship between the yield and sugarcane quality traits during both years of study (Figure 4). During the 2019–2020 period, the unstripped cane yield (USCY) had a strong positive correlation with the stripped cane yield (SCY), millable canes (MC), and number of internodes (NI), consequently showing that these parameters significantly contributed to the USCY. Moreover, the °Brix value had a negative correlation with NI, MC, and SCY, but it showed a positive correlation with the fiber % and pol%. The commercial cane sugar % (CCS%) resulted in a strong positive correlation with the pol%, whereas the cane sugar recovery (CSR%) revealed a strong positive correlation with the pol% and CSS%. Thus, it is indicated that the CSR%, pol%, and CCS% positively correlated to the yield. In addition, during the 2020–2021 period, a similar relationship was exhibited for the yield-related traits. However, the CCS% resulted in a strong negative correlation with the pol%, whereas the CSR% negatively contributed to the pol % and CCS%.

### 2.4. Biplot Analysis

Selected treatments with suitable characters were analyzed using a biplot analysis to evaluate the relationship of the studied traits under subject treatments during the 2019–2020 and 2020–2021 periods (Figure 5). The biplot consisted of four major components, whereas PC–1 and PC–2 reflected 61.7% and 20.4% variations, respectively. The stripped cane yield (SCY), number of internodes (NI), millable canes (MC), and unstripped cane yield were closely related and clustered in one group, and this group showed their performance in the T_7_ (15 irrigations + 110 kg ha^−1^ polymer-coated SSP), T_3_ (18 irrigations + 110 kg ha^−1^ polymer-coated SSP) and T_4_ (18 irrigations + 130 kg ha^−1^ polymer-coated SSP) treatments. However, the cane juice purity (CJP%), commercial cane sugar percentage (CCS%), and cane sugar recovery percentage (CSR%) were clustered in another component by employing the T_8_ (15 irrigations + 130 kg ha^−1^ polymer-coated SSP) and T_6_ (15 irrigations + 90 kg ha^−1^ polymer-coated SSP) treatments, whereas in the same segment, the leading pol% was observed in T_12_ (12 irrigations + 130 kg ha^−1^ polymer-coated SSP) and T_11_ (12 irrigations + 110 kg ha^−1^ polymer-coated SSP). Similarly, the fiber % and °Brix values were documented in the T_9_ (12 irrigations + control) and T_10_ (12 irrigations + 90 kg ha^−1^ polymer-coated SSP) treatments. Moreover, in the results of the 2020–2021 period, PC–1 had the highest (66.5%) variation, and PC–2 resulted in 13.5% variation. In addition, the CSR% and CCS% were clustered on the same axis under the T_10_ (12 irrigations + 90 kg ha^−1^ polymer-coated SSP) treatment. Conversely, the USCY, SCY, and NI were grouped, whereas T_11_ (12 irrigations + 110 kg ha^−1^ polymer-coated SSP) had the highest pol%.

## 3. Discussion

Sugarcane development can be broadly categorized into three stages: the germination, plant establishment, and early tillering phases; grant vegetative growth phase; and maturation and flowering. Numerous studies have focused on managing water stress during the germination, tillering, and vegetative growth stages, as these phases are crucial for crop production. Among these stages, the tillering and stem elongation phases have been found to render sugarcane more susceptible to water stress, affecting both the stem and leaf growth more than the other organs. However, during the maturation phase, moderate water stress can have positive effects on the sucrose yield. This is because photosynthesis is less sensitive to water stress compared to stem growth, which diverts assimilated CO_2_ towards sucrose production and accumulation in the stalk [24,25].

The finding of our study revealed that the maximum number of internodes per cane and the millable cane per square meter were recorded in the treatment where 18 irrigations were applied and 110 kg ha^−1^ of polymer-coated supplements were used. Our finding is similar to the findings of Da Graça [26] and Inman-Bamber [27], as they observed that severe water stress or drought significantly impacts the entire sugarcane plant. The specific morphological and physiological responses of sugarcane plants differ based on various factors such as genotype, stress duration (whether rapid or gradual), stress intensity (severe or mild), and the type of tissue affected [28].

Water stress has a significant impact on both the cane and sugar yield, leading to substantial reductions. Nevertheless, there is promising potential for exploiting genetic variations that could improve the cane and sugar yield under water stress conditions [29]. In response to water stress, sugarcane commonly exhibits various adaptive mechanisms, including leaf rolling, stomatal closure, the inhibition of stalk and leaf growth, leaf senescence, and a reduced leaf area [30]. During water stress, critical growth processes like cell division and cell elongation are interrupted, with stem and leaf elongation being particularly affected [31]. This results in a noticeable reduction in growth. While water deficit influences root development, its impact is relatively less significant compared to the above-ground biomass responses [32].

Sugarcane is a tropical crop with a C_4_ photosynthetic metabolism. When exposed to moderate water stress, certain key physiological changes take place. Notably, there is a decrease in the stomatal conductance (gs), transpiration rate (E), internal CO_2_ concentration (Ci), and photosynthetic rate [33]. These changes are primarily attributed to stomatal limitations, indicating that the reduced water availability leads to the partial closure of the stomata, resulting in a decline in the gas exchange and photosynthetic activity in the plant [34]. Indeed, when sugarcane plants experience mild to moderate dehydration or water stress, the most common initial adaptations are a decrease in the stomatal conductance, transpiration rate, internal CO_2_ concentration, and photosynthetic rate, as mentioned earlier [35]. Additionally, there is a notable inhibition of stalk and leaf growth during this early stage of adaptation to water stress. These responses collectively help the sugarcane plant conserve water and adjust to the reduced water availability in its environment, ensuring better water use efficiency and survival under such conditions [36].

The cane yield is adversely affected due to water scarcity. The finding of this study interpreted that the highest stripped cane yield was obtained during the first year under 18 irrigations, whereas the lowest cane yield was taken where 13 irrigations were applied to sugarcane during the second year, and as far as polymer-coated supplements are concerned, the leading treatment was P2 (where 110 kg ha^−1^ was applied) compared to the other treatments. Our finding is supported by the findings of Robertson [37] and Gentile [38], who depicted that sugarcane is more susceptible to drought stress than flooding stress. Under drought conditions, sugarcane can experience significant yield losses, ranging from 46.2% to 50%. On the other hand, when subjected to flooding, the sugarcane yield can still be adversely affected, with reductions ranging from 14% to 50% compared to well-watered control conditions. Similarly, even in comparison to well-watered conditions, the sugarcane yield can decrease by 18% to 37% when exposed to flooding. These findings highlight the considerable negative impact of both drought and flooding on sugarcane production [39,40]. Moreover, the effects of drought on the morphological parameters will undoubtedly influence the final yield. The cane height and the number of stalks, for instance, exhibit a positive correlation with the cane yield [41]. Therefore, any observed reductions in these parameters are expected to result in a decreased yield. The cane height holds particular significance in determining the yield because it serves as the ultimate sugar sink. Therefore, any reduction in the plant height will invariably lead to a decrease in the yield [42]. Vasantha [43] recorded a substantial 37% reduction in the cane yield under drought conditions, with the sugar yield in the drought treatment being 43.88% lower compared to the control. Interestingly, water deficit during the maturity stage had only marginal effects on the yield. This discrepancy is attributed to the sugarcane plant’s varying water requirements throughout its growth stages, with a higher demand during the early stages compared to the later ones, as emphasized in [44]. Furthermore, De Silva and De Costa [45] noted that drought-affected canes displayed reduced yields across all varieties.

The results regarding the POL, Brix, and commercial cane sugar (CCS) exposed that water scarcity did not affect the POL, °Brix, and CSS, and the maximum value was recorded under water deficit conditions. The results of our findings are supported by Aviva [46], who depicted that the Brix content of sugarcane exposed to drought stress ranged from 17.3% to 18.9%. Remarkably, this demonstrates that even under the challenging conditions of drought stress, the °Brix content of sugarcane remained consistently high. The ability to maintain elevated °Brix levels suggests that these sugarcane varieties possess a certain level of tolerance or resilience to flooding stress, which can be advantageous for maintaining sugar productivity and quality even in adverse growing conditions. Moreover, the sugar accumulation stage is significantly affected by drought stress, and it involves the allocation of additional resources towards the synthesis and storage of sucrose. As a consequence, there is a potential increase in cane sugar, as the plant boosts its sucrose production to conserve water. In this scenario, drought stress can yield a positive effect on the cane sugar percentage, thereby enhancing its quality. However, the overall study’s results show similar findings as those described in [47]. Overall, several agronomic and modern biotechnological approaches should be implanted through methodological measures to protect sugarcane plants aginst various biotic and abiotic stressors [48,49,50,51,52,53,54]. In addition, the findings of this study show that 110 kg ha^−1^ of polymer-coated SSP considerably increased the sugarcane yield along with 18 irrigations.

## 4. Materials and Methods

### 4.1. Field Preparation and Trial Execution

Two years (2019–2020 and 2020–2021 periods) of field trials were executed in Faisalabad to evaluate the response of polymer-coated SSP under varying irrigation regimes. Field trials were set up during the third week of February in both years of study at PARS (Post Graduate Agriculture Research Station (31°24′04′′ N, 73°01′55 E and altitude 184 m)) at the University of Agriculture Faisalabad. Experiments were conducted using a randomized complete block design with a split-plot arrangement with three replications. CFP-249 sugarcane cultivar was exposed to three water regimes (18, 15, and 12 irrigations, respectively) and four levels of polymer-coated SSP (0 kg ha^−1^, 90 kg ha^−1^, 110 kg ha^−1^, and 130 kg ha^−1^). A single irrigation was applied at the rate of 4 acre inches in each water regime by employing the surface irrigation method, and the total irrigation amount was applied at 72 acre inches (18 irrigations), 60 acre inches (15 irrigations), and 48 acre inches (12 irrigations) during the whole crop period. The crop was sown in 120 cm wide trenches, which were made by sugarcane planters. Two-eyed bud setts were placed in trenches and covered with one inch of soil for better emergence. Moreover, 90 kg ha^−1^, 110 kg ha^−1^, and 130 kg ha^−1^ polymer-coated SSP regimes were applied at the time of cane sett placement, whereas 170 kg ha^−1^ nitrogen was applied in splits, and 60 kg ha^−1^ potassium was applied at the time of sowing. In addition, varying water regimes were applied through the skip irrigation method, and other cultural practices including hoeing, weeding, and insect control measures were performed according to the requirement. Before plantation, soil sampling was performed at a 0–30 cm depth with the help of soil auger, and the soil analysis data are given in Table 3. Moreover, the weather attributes of this study were collected from the Metrological Cell University of Agriculture Faisalabad (Figure 6).

### 4.2. Data Collection

Both years’ studied data were collected in 1st week of March. Data were collected during both years’ experimentation, which included the traits of number of internodes per cane, millable canes (m^−2^), stripped cane yield, and unstripped cane yield as well as sugarcane quality traits.

#### Sugarcane Juice Extraction

For extraction of sugarcane, the cleaned sugarcane was cut into small pieces using a cutter. These prepared sugarcane pieces were then directed to the rolling mill, where they underwent pressing to extract the juices. After the initial pressing, hot water was reintroduced to further extract the maximum percentage of juice from the compressed sugarcane. The obtained juice was subsequently sent for purification through juice clarifiers and quality traits were measured [55]. After the measurement of juice parameters, subsequent traits were calculated using the procedure below.

Commercial cane sugar (CCS) percentage was examined using the protocols of [49].
$$CCS\%=\frac{3}{2}\;P\;\left(1-\frac{F+5}{100}\right)-\frac{1}{2}B\;\left(1-\frac{F+3}{100}\right)$$

Here, P = Pol% in juice, which shows the presence of sucrose content, B = °Brix, which shows the presence of total soluble solids, and F = fiber%

In addition, the cane sugar recovery (CSR) percentage was determined using the following equation [56]:CSR% = CCS% × 0.94

However, the cane juice purity (CJP) percentage was calculated using the following formula:CJP% = (Pol %)/(°Brix) × 100

### 4.3. Statistical Analysis

Statistical analyses were performed for collected traits using analysis of variance technique, and treatment means were differentiated by employing the LSD test at a 95% confidence interval. Moreover, graphs of interactive effects were made using the paired comparison plot technique, and analyses were accomplished using OriginPro–2022 software (Originlab, Northampton, MA, USA). Pearson correlation analysis was executed by adopting two-tailed tests (df–2), and biplot analysis was also conducted using the aforementioned software.

## 5. Conclusions

Concluding the results of this study, drought stress exerts a significant impact on various parameters of sugarcane, including the millable cane yield (m^−2^), number of internodes per cane, stripped cane yield, and unstripped cane yield. However, the pol%, Brix, and commercial cane sugar percentages were found to be significantly improved under limited water conditions. In addition, the application of 110 kg ha^−1^ polymer-coated SSP proved to be a promising approach, enhancing the cane yield despite the challenges posed by drought. Moreover, the application of 130 kg ha^−1^ of polymer-coated SSP along with a limited water supply to the sugarcane significantly improved the pol%, cane juice purity %, commercial cane sugar %, and cane sugar recovery %. This finding highlights the potential of using polymer-coated supplements as a viable strategy to mitigate the adverse effects of drought on sugarcane production. 

## Figures and Tables

**Figure 1 plants-12-03432-f001:**
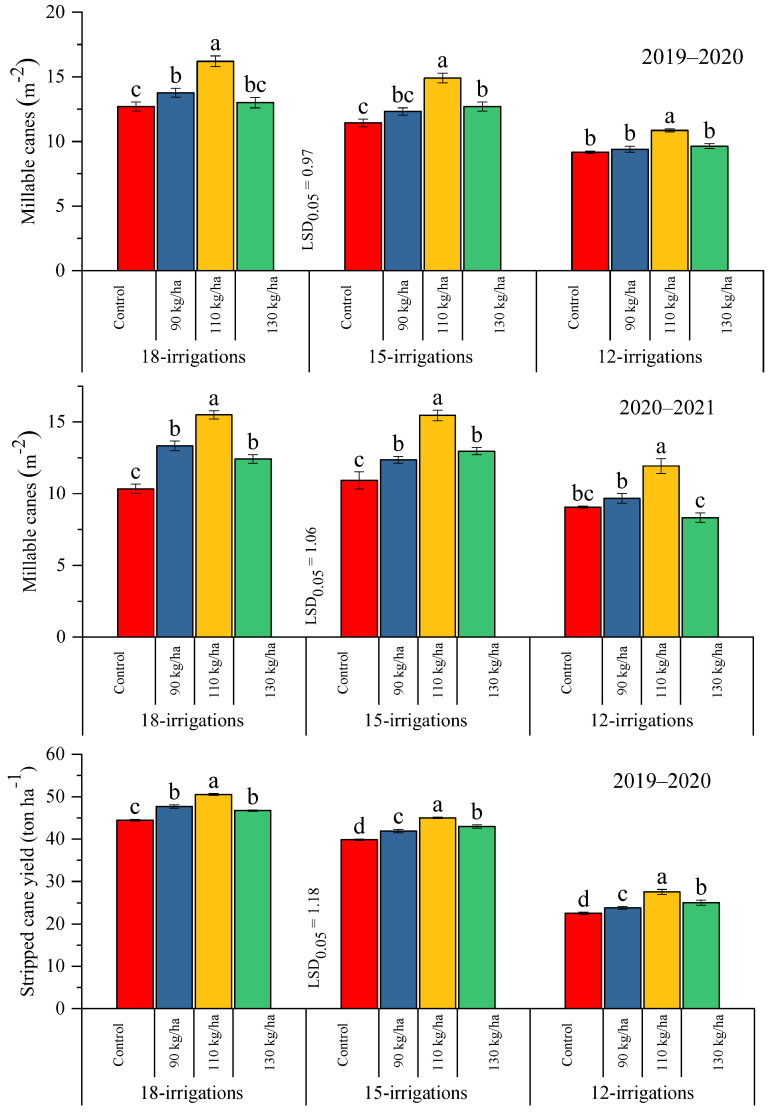
Effect of different doses of polymer-coated SSP on millable canes and stripped cane yield under differential water regimes across the years. Different letters on bars showed significant variation among studied traits.

**Figure 2 plants-12-03432-f002:**
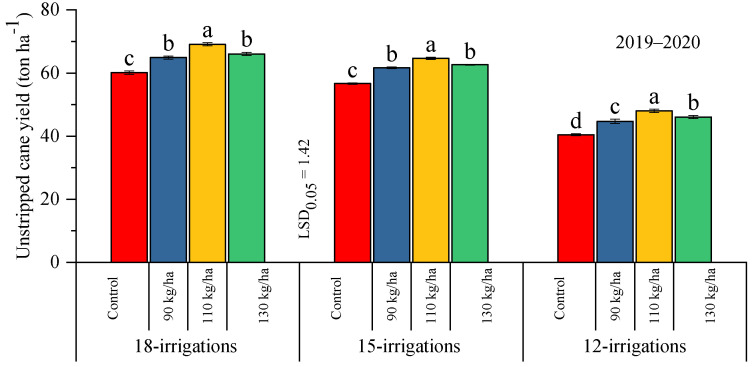
Effect of different doses of polymer-coated SSP on unstripped cane yield and °brix value under differential water regimes across the years. Different letters on bars show significant variation among studied traits.

**Figure 3 plants-12-03432-f003:**
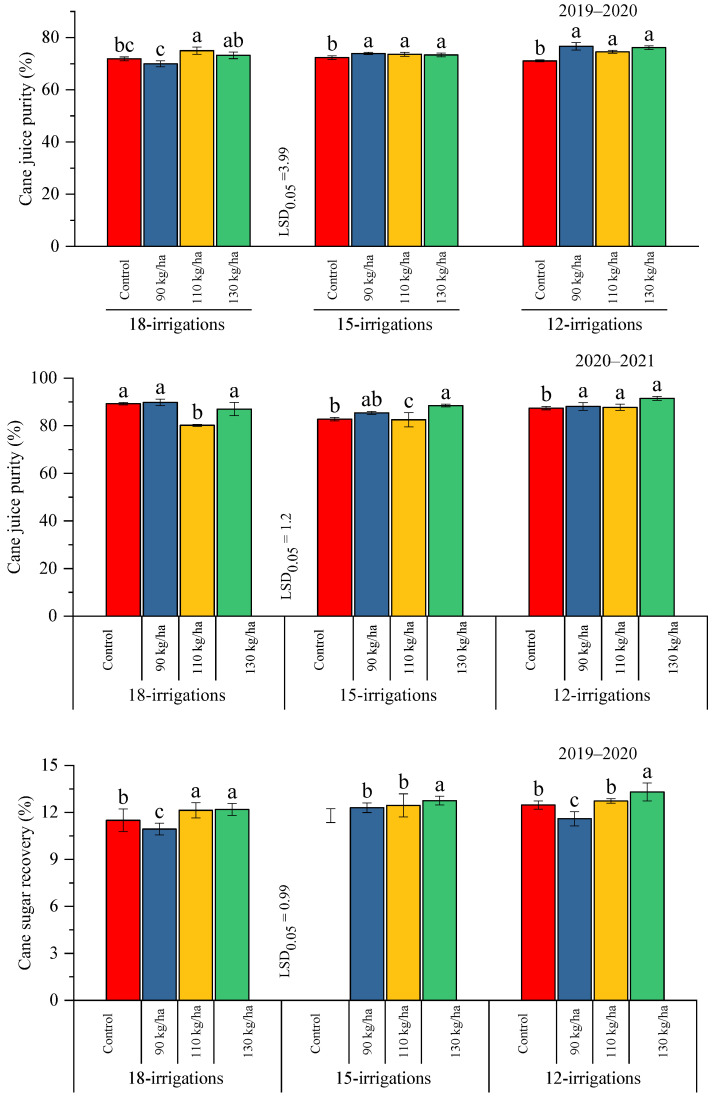
Effect of different doses of polymer-coated SSP on cane juice purity % and cane sugar recovery % under differential water regimes across the years. Different letters on bars show significant variation among studied traits.

**Figure 4 plants-12-03432-f004:**
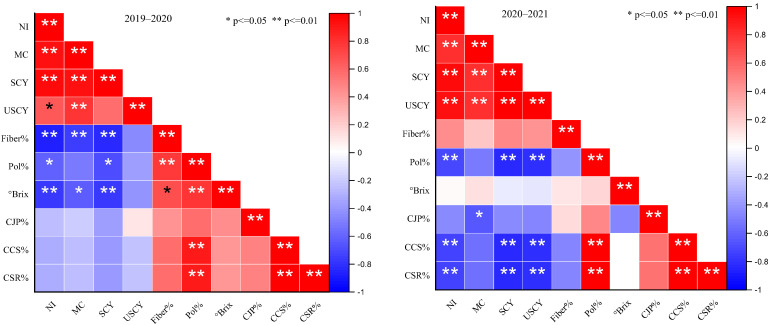
Correlation analyses between yield component and sugar-related traits under varying doses of polymer-coated supplement and irrigation regimes during the 2019–2020 and 2020–2021 periods. Here, NI = several internodes cane^−1^, MC = millable canes (m^−2^), SCY = stripped cane yield, USCY = unstripped cane yield, CJP% = cane juice purity%, CCS% = commercial cane sugar %, and CSR% = cane sugar recovery %.

**Figure 5 plants-12-03432-f005:**
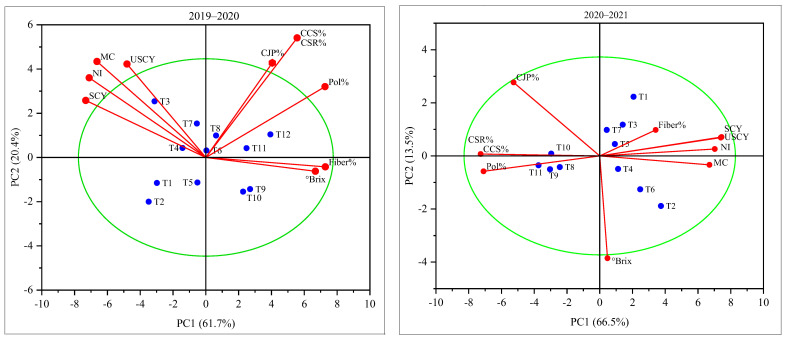
Principal component analysis yield component and sugar-related traits under varying doses of polymer-coated supplement and irrigation regimes during the 2019–2020 and 2020–2021 periods.

**Figure 6 plants-12-03432-f006:**
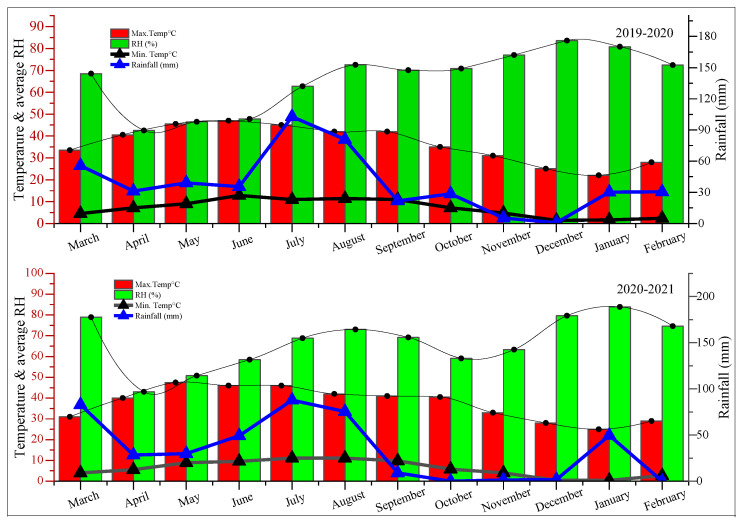
Weather data during the length of field experiments over the years.

**Table 1 plants-12-03432-t001:** Effect of biodegradable polymer on sugarcane yield components and fiber % under differential irrigation regimes across the years. Similar letters represent an insignificant (*p* ≥ 0.05) difference among the subject treatments, and the ± symbol denotes the standard error of treatments.

Treatments	Parameters									
	No. of Internodes Cane^−1^		Millable Canes m^−2^		Stripped Cane Yield t ha^−1^		Unstripped Cane Yield t ha^−1^		Fiber %	
	2019–2020	2020–2021	2019–2020	2020–2021	2019–2020	2020–2021	2019–2020	2020–2021	2019–2020	2020–2021
W_1_ = 18 irrigations	16.78 ± 0.25 ^a^	15.00 ± 0.55 ^a^	13.92 ± 0.37 ^a^	12.90 ± 0.31 ^a^	47.34 ± 0.25 ^a^	41.65 ± 0.42 ^a^	65.05 ± 0.50 ^a^	59.26 ± 0.53 ^a^	11.96 ± 0.11	12.97 ± 0.38
W_2_ = 15 irrigations	14.49 ± 0.34 ^b^	13.02 ± 0.62 ^b^	12.84 ± 0.32 ^b^	12.93 ± 0.35 ^a^	42.42 ± 0.27 ^b^	37.68 ± 0.28 ^b^	61.42 ± 0.24 ^b^	56.32 ± 0.27 ^b^	12.49 ± 0.30	12.59 ± 0.24
W_3_ = 12 irrigations	10.85 ± 0.16 ^c^	10.17 ± 0.31 ^c^	9.76 ± 0.15 ^c^	9.75 ± 0.31 ^b^	24.73 ± 0.44 ^c^	22.96 ± 0.55 ^c^	44.81 ± 0.51 ^c^	40.18 ± 0.52 ^c^	12.80 ± 0.46	12.55 ± 0.34
LSD (α 5%)	0.24	0.59	0.45	0.71	0.41	1.36	0.52	0.34	NS	NS
P_0_ = control	13.29 ± 0.19 ^c^	11.96 ± 0.37 ^c^	11.10 ± 0.24 ^c^	10.11 ± 0.32 ^c^	35.59 ± 0.21 ^c^	31.39 ± 0.29 ^c^	52.41 ± 0.36 ^d^	48.84 ± 0.16 ^c^	12.47 ± 0.33	12.68 ± 0.18
P_1_ = 90 kg ha^−1^	13.18 ± 0.31 ^c^	11.97 ± 0.48 ^c^	11.83 ± 0.28 ^b^	11.79 ± 0.30 ^b^	37.79 ± 0.38 ^b^	33.82 ± 0.41 ^b^	57.09 ± 0.46 ^c^	51.74 ± 0.50 ^b^	12.42 ± 0.35	12.84 ± 0.49
P_2_ = 110 kg ha^−1^	15.55 ± 0.31 ^a^	14.02 ± 0.60 ^a^	13.99 ± 0.30 ^a^	14.30 ± 0.39 ^a^	41.03 ± 0.32 ^a^	36.89 ± 0.45 ^a^	60.61 ± 0.47 ^a^	54.77 ± 0.56 ^a^	12.31 ± 0.24	12.62 ± 0.31
P_3_ = 130 kg ha^−1^	14.14 ± 0.21 ^b^	12.97 ± 0.52 ^b^	11.78 ± 0.31 ^b^	11.24 ± 0.28 ^b^	38.24 ± 0.37 ^b^	34.28 ± 0.52 ^b^	58.27 ± 0.38 ^b^	52.32 ± 0.53 ^b^	12.48 ± 0.23	12.66 ± 0.30
LSD (α 5%)	0.53	0.89	0.56	0.61	0.68	0.72	0.82	0.86	NS	NS
Interaction	NS	NS	*	**	*	**	*	*	NS	NS

Here, * = *p* ≤ 0.05 and ** = *p* ≤ 0.01, W_X_ = irrigation regimes, and P_X_ = polymer-coated SSP regimes.

**Table 2 plants-12-03432-t002:** Effect of biodegradable polymer on sugarcane quality traits under differential irrigation regimes across the years. Similar letters represent an insignificant (*p* ≥ 0.05) difference among the subject treatments, and the ± symbol denotes the standard error of treatments.

Treatments	Parameters									
	Pol%		°Brix		CCS%		CJP%		CSR%	
	2019–2020	2020–2021	2019–2020	2020–2021	2019–2020	2020–2021	2019–2020	2020–2021	2019–2020	2020–2021
W_1_ = 18 irrigations	16.15 ± 0.30 ^c^	16.08 ± 0.18 ^b^	18.05 ± 0.40 ^c^	19.01 ± 0.34	12.44 ± 0.52 ^b^	11.71 ± 0.25 ^b^	72.48 ± 1.13	86.56 ± 1.26 ^ab^	11.69 ± 0.49 ^b^	11.00 ± 0.23 ^b^
W_2_ = 15 irrigations	17.13 ± 0.36 ^b^	16.32 ± 0.13 ^b^	19.13 ± 0.15 ^b^	19.27 ± 0.24	13.12 ± 0.46 ^ab^	12.04 ± 0.16 ^b^	73.28 ± 0.63	84.77 ± 1.22 ^b^	12.33 ± 0.44 ^ab^	11.32 ± 0.15 ^b^
W_3_ = 12 irrigations	17.80 ± 0.28 ^a^	17.83 ± 0.14 ^a^	20.47 ± 0.35 ^a^	19.74 ± 0.35	13.33 ± 0.38 ^a^	13.85 ± 0.18 ^a^	74.60 ± 0.78	88.67 ± 1.13 ^a^	12.53 ± 0.36 ^a^	13.02 ± 0.17 ^a^
LSD (α 5%)	0.60	0.39	0.40	NS	0.80	0.60	NS	3.62	0.75	0.56
P_0_ = control	16.60 ± 0.34 ^b^	16.26 ± 0.20 ^c^	18.59 ± 0.25 ^b^	19.13 ± 0.28 ^b^	12.69 ± 0.50 ^a,b^	11.96 ± 0.19 ^c^	71.74 ± 0.61 ^b^	86.45 ± 0.64 ^b^	11.92 ± 0.47 ^a,b^	11.25 ± 0.18 ^c^
P_1_ = 90 kg ha^−1^	16.64 ± 0.34 ^b^	16.70 ± 0.10 ^b^	19.52 ± 0.24 ^a^	19.12 ± 0.30 ^b^	12.36 ± 0.40 ^b^	12.55 ± 0.12 ^b^	73.49 ± 0.98 ^a^	87.77 ± 1.20 ^a,b^	11.62 ± 0.37 ^b^	11.80 ± 0.11 ^b^
P_2_ = 110 kg ha^−1^	17.32 ± 0.30 ^a^	16.84 ± 0.13 ^b^	19.48 ± 0.41 ^a^	19.90 ± 0.46 ^a^	13.24 ± 0.48 ^a,b^	12.53 ± 0.21 ^b^	74.36 ± 0.90 ^a^	83.46 ± 1.57 ^c^	12.44 ± 0.45 ^a,b^	11.78 ± 0.20 ^b^
P_3_ = 130 kg ha^−1^	17.54 ± 0.26 ^a^	17.16 ± 0.16 ^a^	19.28 ± 0.28 ^a^	19.21 ± 0.21 ^b^	13.57 ± 0.43 ^a^	13.09 ± 0.27 ^a^	74.21 ± 0.90 ^a^	88.98 ± 1.40 ^a^	12.75 ± 0.41 ^a^	12.30 ± 0.25 ^a^
LSD (α 5%)	0.61	0.27	0.62	0.52	0.89	0.37	1.45	2.44	0.84	0.34
Interaction	NS	NS	*	*	NS	NS	**	*	*	*

Here; W_X_ = irrigation regimes, and P_X_ = polymer-coated SSP regimes, whereas CCS% = commercial cane sugar %, CJP% = cane juice purity %, CSR = cane sugar recovery%, * = *p* ≤ 0.05, and ** = *p* ≤ 0.01.

**Table 3 plants-12-03432-t003:** Physio-chemical profile of soil of 2019–2020 and 2020–2021 periods.

Soil Characters	2019–2020		2020–2021	
	Units	Value	Units	Value
Texture	Loam	–	–	–
pH	–	7.5 ± 0.1	–	7.54 ± 0.1
EC	d S L^–1^	1.87 ± 0.02	d S L^–1^	1.93 ± 0.04
Organic matter	µg g^–1^	6.19 ± 0.19	µg g^–1^	6.87 ± 0.14
Nitrogen	µg g^–1^	292 ± 8.16	µg g^–1^	324 ± 11.13
Phosphorus	µg g^–1^	7.89 ± 0.28	µg g^–1^	8.45 ± 0.31
Potassium	µg g^–1^	235 ± 17	µg g^–1^	256 ± 10.11

## Data Availability

All data generated or analyzed during this study are included in this published article.

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
