# Peer review of "Enhancing Sugarcane Yield and Sugar Quality through Optimal Application of Polymer-Coated Single Super Phosphate and Irrigation Management"

_plants, 2023, doi:10.3390/plants12193432_

Round 1

Reviewer 1 Report

I am wndering that the authors said that drought stress had very little effect on cane quality traits. From the data in Table 2, however, the least irrigation treatment significantly had the highest cane quality with the best quality traits. It is a general concept that appropriate drought condition during sugar accumulation stage can improve cane quality (sugar related traits). I suggest the authors imp[oved the related descriptions in the manuscript.

The units of some parameters should be standarized. For example, the unit of Brix, we use oBx instead of %.

There are still some concept mistakes and typos in the contexts.

Author Response

I am wondering that the authors said that drought stress had very little effect on cane quality traits. From the data in Table 2, however, the least irrigation treatment significantly had the highest cane quality with the best quality traits. It is a general concept that appropriate drought condition during sugar accumulation stage can improve cane quality (sugar related traits)?

Response: Drought stress during the sugar accumulation stage can trigger a response in sugar cane plants to allocate more resources to the synthesis and storage of sucrose, the primary sugar of interest. This can result in an increase in cane sugar% as the plant produces more sucrose to conserve water. In this case, drought can have a positive effect on cane sugar %, leading to better quality.

The units of some parameters should be standardized. For example, the unit of Brix, we use oBx instead of %.

Response: Unit of brix% is replaced with °Brix throughout the manuscript.

I suggest the authors improved the related descriptions in the manuscript. There are still some concept mistakes and typos in the contexts.

Response: We are thankful to the respected reviewer. Needful is done.

Reviewer 2 Report

The manuscript authored by Sajid et al. titled "Enhancing Sugarcane Yield and Sugar Quality through Optimal Application of Polymer Coated SSP and Irrigation Management" is relatively straightforward, but it presents certain limitations. The depth of analysis in the study appears to be quite limited. Additionally, the novelty of the research is constrained due to a previously published study by the same author, titled "Agronomic Responses of Sugarcane (Saccharum officinarum L.) Ratoon to Natural and Synthetic Supplements under Water Deficit Conditions," which covers similar ground

One area of concern is the lack of clarity regarding the rationale behind selecting different levels of polymer coated SSP (single super phosphate). The prior research on polymer coated SSP is inadequately introduced, making it difficult for readers to grasp its relevance to the current studyThe methodology section requires substantial improvement. It would greatly benefit from the inclusion of more comprehensive details about the field experiments, including information on biological replicates, the application of treatments, and the process of juice extraction. These details are vital for the transparency and reproducibility of the research. Furthermore, the discussion section appears overly descriptive and includes several introductory sentences. It lacks a mechanistic understanding of the main results, which is essential for providing meaningful insights and interpretations. Finally, the manuscript needs an improvement at writing level, for example 

In the abstract (Therefore, two years field experimentation was executed to assess the potential of varying polymer coated SSP regimes under differential water regimes) potential for what?

CFP-249 sugarcane cultivar was exposed to under three water levels, need correction 

Minor editing of English language required

Author Response

The manuscript authored by Sajid et al. titled "Enhancing Sugarcane Yield and Sugar Quality through Optimal Application of Polymer Coated SSP and Irrigation Management" is relatively straightforward, but it presents certain limitations. The depth of analysis in the study appears to be quite limited. Additionally, the novelty of the research is constrained due to a previously published study by the same author, titled "Agronomic Responses of Sugarcane (Saccharum officinarum) Ratoon to Natural and Synthetic Supplements under Water Deficit Conditions," which covers similar ground. One area of concern is the lack of clarity regarding the rationale behind selecting different levels of polymer coated SSP (single super phosphate). The prior research on polymer coated SSP is inadequately introduced, making it difficult for readers to grasp its relevance to the current study.

Response: Respected reviewer, thank you for your comment. Regarding the issue of novelty, it's important to differentiate between the two studies. While the previous paper focused on natural and synthetic supplements under water deficit conditions, the recent manuscript titled "Enhancing Sugarcane Yield and Sugar Quality through Optimal Application of Polymer Coated SSP and Irrigation Management" takes a distinct approach by exploring the effects of different doses of single super phosphate (SSP) and optimizing the application of polymer-coated SSP. The shift in focus from supplements to the application of SSP, specifically polymer-coated SSP, is a significant departure from the prior research. This change in emphasis allows the recent manuscript to investigate a different avenue for enhancing sugarcane yield and sugar quality. Polymer-coated SSP represents a unique and potentially more efficient method for nutrient delivery and irrigation management, and its effects on sugarcane growth and quality deserve attention. By emphasizing this shift in research focus and highlighting the specific contributions and advantages of studying polymer-coated SSP in combination with irrigation management, the recent manuscript can be seen as building upon the previous work rather than replicating it. It offers valuable insights into a distinct approach to address similar challenges in sugarcane cultivation.

The methodology section requires substantial improvement. It would greatly benefit from the inclusion of more comprehensive details about the field experiments, including information on biological replicates, the application of treatments, and the process of juice extraction. These details are vital for the transparency and reproducibility of the research.

Response: Methodology sections have been improved as suggested by reviewers and highlighted in the manuscript. Application methods of polymer-coated single super phosphate and as well as cane quality parameter extraction like juice extraction methods were incorporated.

Furthermore, the discussion section appears overly descriptive and includes several introductory sentences. It lacks a mechanistic understanding of the main results, which is essential for providing meaningful insights and interpretations.

Response: Needful is done as per suggestion.

Finally, the manuscript needs an improvement at writing level, for example In the abstract (Therefore, two years field experimentation was executed to assess the potential of varying polymer coated SSP regimes under differential water regimes) potential for what? CFP-249 sugarcane cultivar was exposed to under three water levels, need correction

Response: Needful is done.

Reviewer 3 Report

Manuscript plants-2576663 Enhancing Sugarcane Yield and Sugar Quality through Optimal Application of Polymer Coated SSP and Irrigation Management”

​The authors conducted a 2-year field experiment to assess whether single super phosphate SSP applied in different doses and 3 levels of irrigation treatments  would have a positive impact on yield and quality of sugarcane. The effects of the experiment factors were tested probably somewhere in Pakistan, however, the authors did not mention the coordinates of the location.

The topic of the article is an interesting and valuable topic, however, the manuscript needs some revision. 

The explanation of the abbreviation SSP should be on line 20 when it first appears

Please provide information regarding the location of the experimental field.

There is no information describing the climate condition of the site. What is more, no weather conditions during the two periods of sugarcane grooving is presented.

How much is the sugar cane's water consumption, did irrigation cover this consumption or was it deficient? If it was deficit irrigation, please specify how many % of evapotranspiration  were covered.

There is no information on how much water was used for irrigation. Irrigation levels mean nothing to a potential reader! Please enter absolute data. What guided the setting of irrigation levels? evapotranspiration? How was evapotranspiration calculated? Where did the data for the calculations come from - give the coordinates.

What type of irrigation was used?

What was the single dose?

How often has it been used? What were the irrigation dates based on?

Were they distributed regularly over time, or differentiated at different stages of plant development?

Table 1 . Describe the symbols W and P in the table. Not mentioned in the text or in the table heading.

L. 154. 13 irrigation treatments? Please revise it.

L. 185. What does it mean :Moreover, different treatments of polymer coated SSP was used at 185 the time of plantation ..." . The treatments have to be described in details.

The section results seems to be chaotic, when separately are described the results obtained in the particular years, characterized by great variation. There is a need od synthesis of the results first, then analysis of differences and explanations why the different results were obtained.

The discussion of results obtained in the experiment should be deepened.

The Conclusion is very laconic.

The English needs to be checked by a native speaker.

After the revision, I recommend the manuscript be published in the journal.

The manuscript needs revision especially in the section of Materials and Methods.

Author Response

The authors conducted a 2-year field experiment to assess whether single super phosphate SSP applied in different doses and 3 levels of irrigation treatments would have a positive impact on yield and quality of sugarcane. The effects of the experiment factors were tested probably somewhere in Pakistan; however, the authors did not mention the coordinates of the location.

Response: The location of experimental site has been added.

The explanation of the abbreviation SSP should be on line 20 when it first appears

Response: SSP stands for single super phosphate and it was also incorporated in line 20.

Please provide information regarding the location of the experimental field.

Response: Needful is done.

There is no information describing the climate condition of the site. What is more, no weather conditions during the two periods of sugarcane grooving is presented.

Response: A graph of weather data has been added.

How much is the sugar cane's water consumption, did irrigation cover this consumption or was it deficient? If it was deficit irrigation, please specify how many % of evapotranspiration was covered.

Response: The water needs of sugarcane can fluctuate based on a range of factors, including the climate, soil type, growth stage, and local circumstances. Sugarcane is known to be a crop with relatively high water demands, particularly during crucial growth phases like tillering and sugar accumulation. In a typical crop cycle, sugarcane generally requires 64 acre-inches of water. However, due to increasing temperatures, we applied two additional irrigations, totaling 72 acre-inches of irrigation water for one treatment. It's worth noting that we did not calculate the evapotranspiration in our study since it was not a mandatory component of our research focus.

There is no information on how much water was used for irrigation. Irrigation levels mean nothing to a potential reader! Please enter absolute data. What guided the setting of irrigation levels? evapotranspiration? How was evapotranspiration calculated? Where did the data for the calculations come from - give the coordinates.

Response: Thank you for the comment, we have used 4 acres inches of water for each irrigation, one acre inch is equal to 25 mm of water.

What type of irrigation was used?

Response: Surface irrigation was applied

What was the single dose?

Response: Single irrigation of 4 acre inch was applied

How often has it been used? What were the irrigation dates based on? Were they distributed regularly over time, or differentiated at different stages of plant development?

Response: For drought skip irrigation method was used and the irrigation schedule was based on growth stages like emergence, tillering, grand growth etc.

Table 1. Describe the symbols W and P in the table. Not mentioned in the text or in the table heading.

Response: Needful is done

L. 154. 13 irrigation treatments? Please revise it.

Response: Needful has been done

L. 185. What does it mean: Moreover, different treatments of polymer coated SSP was used at 185 the time of plantation ...”. The treatments have to be described in details.

Response: Needful has been done

The section results seem to be chaotic, when separately are described, the results obtained in the particular years, characterized by great variation. There is a need od synthesis of the results first, then analysis of differences and explanations why the different results were obtained.

Response: The results section has been revised

The discussion of results obtained in the experiment should be deepened.

Response: The discussion section has been improved according to the suggestion of respected reviewer.

The Conclusion is very laconic.

Response: The conclusion section has been corrected

Round 2

Reviewer 1 Report

No

Reviewer 2 Report

The manuscript is improved and can be published in current status 

Minor editing of English language required

Reviewer 3 Report

Thank you the Authors for the corrections  and reviewig   the manuscript.